# A Comparison Study of Marginal and Internal Fit Assessment Methods for Fixed Dental Prostheses

**DOI:** 10.3390/jcm8060785

**Published:** 2019-06-01

**Authors:** Keunbada Son, Sangbong Lee, Seok Hyon Kang, Jaeseok Park, Kyu-Bok Lee, Mansik Jeon, Byoung-Ju Yun

**Affiliations:** 1Department of Dental Science, Graduate School, Kyungpook National University, 2177 Dalgubeol-daero, Jung-gu, Daegu 41940, Korea; sonkeunbada@gmail.com; 2Advanced Dental Device Development Institute (A3DI), Kyungpook National University, 2177 Dalgubeol-daero, Jung-gu, Daegu 41940, Korea; sehykase@knu.ac.kr; 3College of IT Engineering, School of Electronics Engineering, Kyungpook National University, 80, Daehak-ro, Buk-gu, Daegu 41566, Korea; emg0218@knu.ac.kr (S.L.); pjs0307zz@knu.ac.kr (J.P.); 4Department of Prosthodontics, School of Dentistry, Kyungpook National University, 2177 Dalgubeol-daero, Jung-gu, Daegu 41940, Korea

**Keywords:** marginal and internal fit, dentistry, dental prosthesis, zirconia crown, micro-computed tomography, optical coherence tomography

## Abstract

Numerous studies have previously evaluated the marginal and internal fit of fixed prostheses; however, few reports have performed an objective comparison of the various methods used for their assessment. The purpose of this study was to compare five marginal and internal fit assessment methods for fixed prostheses. A specially designed sample was used to measure the marginal and internal fit of the prosthesis according to the cross-sectional method (CSM), silicone replica technique (SRT), triple scan method (TSM), micro-computed tomography (MCT), and optical coherence tomography (OCT). The five methods showed significant differences in the four regions that were assessed (*p* < 0.001). The marginal, axial, angle, and occlusal regions showed low mean values: CSM (23.2 µm), TSM (56.3 µm), MCT (84.3 µm), and MCT (102.6 µm), respectively. The marginal fit for each method was in the range of 23.2–83.4 µm and internal fit (axial, angle, and occlusal) ranged from 44.8–95.9 µm, 84.3–128.6 µm, and 102.6–140.5 µm, respectively. The marginal and internal fit showed significant differences depending on the method. Even if the assessment values of the marginal and internal fit are found to be in the allowable clinical range, the differences in the values according to the method should be considered.

## 1. Introduction

The marginal fit of a fixed prosthesis is one of the most important factors for successful prosthetic treatment [1,2]. An ideal marginal fit maintains a healthy periodontal status and prevents cement dissolution [3,4]. On the other hand, a poor marginal fit has a negative impact on the periodontium, making it difficult to perform long-term maintenance of the patient’s health following implant placement [5,6,7]. In addition, an excellent internal fit increases the retention of the prosthesis [8]. For these reasons, numerous studies have been conducted on the marginal and internal fit of a prosthesis to determine its prognosis [1,2,3,4,5,6,7,8,9,10,11,12,13,14,15,16,17,18,19,20,21,22,23,24,25,26,27,28,29,30,31,32,33,34,35,36,37,38].

Previous studies have discussed the pros and cons of the methods introduced for assessing the marginal and internal fit. The pros and cons of the five types of assessment methods are as follows:(1)The cross-sectional method (CSM): the CSM is a method in which after cementation, the desired part of the prosthesis is cut and measured with an optical or electronic microscope. Since the actual prosthesis is cut and measured, it has the advantage of allowing accurate measurement of the internal and marginal fit of the prosthesis. However, the disadvantages are that the method requires the samples to be destroyed, and the measurements cannot be made directly in the oral cavity [11,12,29].(2)The silicone replica technique (SRT): SRT is performed using the same protocol as cementation of a prosthesis. However, the method involves injecting silicone instead of cement inside the prosthesis and duplicating the internal and marginal fit for measurement. Since this is a relatively simple, low-cost method allowing the measurements to be made directly in the oral cavity, it has been utilized in many studies [21,22,23,24,25,26,30]. However, there is a possibility of deformation and tearing of the impression materials [31,32,33,34]. Additionally, CSM and SRT can make assessments only using two-dimensional (2D) analysis.(3)The triple scan method (TSM): the TSM is a method of scanning the internal and external aspects of the prosthesis, the abutment tooth, and the prosthesis at the try-in stage in order to obtain three-dimensional (3D) data and measure the marginal and internal fit by overlapping the 3D data on an analysis software. It is a non-destructive, non-radioactive method capable of providing reproducible results at any time by scanning the data. However, miscalculations may occur due to possible inaccuracy and overlapping of the scanned data [27,28,34,35].(4)Micro-computed tomography (MCT): MCT measures the internal and marginal fit of the prosthesis using radiography. The merits of this method include a high resolution and the ability to measure the desired parts by obtaining 3D images. However, the disadvantages of this method include the difficulty to measure metallic prostheses due to the presence of artifacts and an increased risk of exposure to radiation [3,4,13,14,15,21,36].(5)Optical coherence tomography (OCT): OCT is a method of measurement using higher resolution 2D or 3D images in optical scattering media using coherent light. This is a non-destructive, non-radiological method with the advantage of allowing the acquisition of higher resolution images in real time, which are often utilized for in vivo research [37]. On the other hand, its disadvantage includes difficulty in measuring very thick or optical-opaque materials [16,17,18,19,20,21]. Both 2D and 3D analysis are possible with TSM, MCT, and OCT.

Previous studies have compared the accuracy of methods for analyzing the marginal and internal fit measurements of fixed prostheses [14,18,21,34,39,40]. However, since the objective of the previous studies was to increase the reliability of the assessment method, they only compared two methods to identify an appropriate approach, rather than comparing the various existing methods.

Also, the studies by Han et al. [21], Boitelle et al. [39], and Zeller et al. [40] reported different findings even when comparing only two methods. Additionally, the findings from previous studies examining the various methods for measuring the marginal and internal fit of the prosthesis were difficult to compare due to the differences in the experimental condition in each study. Thus, although previous studies have examined the significance of the various assessment methods, an objective comparison between them is still lacking.

With the recent advances in computer-aided design and computer-aided manufacturing (CAD-CAM) systems, the marginal gap of the prosthesis is a very low value within 100 µm, while most of the literature reports the clinically allowable marginal gap to be within a range of 100–120 µm [11,41,42,43,44]. Several previous studies have assessed the fitness of fixed prostheses based on this clinically allowable range [1,41,45,46,47], but if there are differences in the resulting values based on the assessment method used, an objective comparison is difficult. Thus, for precise measurement, it is necessary to verify the differences between the various methods recently presented.

In this study, in order to verify the differences through an objective comparison of five evaluation methods, the marginal and internal fit were measured in cross-sectional images obtained from the sample under identical conditions. This study aimed to conduct a comparative assessment of the marginal and internal fit methods introduced in previous studies (CSM, SRT, TSM, MCT, and OCT). A null hypothesis was set as follows: there would be no differences in the marginal and internal (axial, angle, and occlusal region) fit measured according to the CSM, SRT, TSM, MCT, and OCT assessment methods.

## 2. Materials and Methods

### 2.1. Sample Preparation

The study model and the guide template were specially designed for assessing the marginal and internal fit of the prosthesis (Figure 1). In the dental model (ANA-4; Frasaco GmbH, Tettnang, Germany), the upper right first molar was prepared as follows: 1.5 mm reduction of the occlusal surface, 1 mm reduction of the axial wall, a 1.2 mm chamfer margin, and a 6-degree vertical angle. The study model was scanned using a model scanner (FREEDOMHD; DOF, Seoul, Korea) to obtain a virtual model. This virtual model was loaded in the computer-aided design (CAD) software (SolidWorks 2014 software; Dassault Systems Solid-Works Corp., Waltham, MA, USA) and designed so that it could be combined with the analysis guide template at the bottom of the abutment tooth (Figure 1a). The analysis guide template was designed to be a fixed 0.5 mm gap passing through the center of the mesiodistal and buccolingual surfaces of the abutment tooth at the bottom of the abutment tooth in the CAD software (Figure 1a), allowing the analysis to be conducted on the same plane in all five methods (Figure 1b,c). The designed study model and analysis guide were then fabricated using a 3D printer (ZENITH; Dentis, Daegu, Korea) with a 16 µm layer (Figure 2). Resins were used for 3D printing the model (ZMD-1000B, ZMD0171208B02; Dentis, Daegu, Korea), as well as the guide (ZMD-1000B, ZCL0170621A06; Dentis, Daegu, Korea).

The fabricated study model was scanned using a model scanner, and using dental CAD software (Exocad; Exocad GmbH, Darmstadt, Germany) an upper right first molar coping was designed with the following parameters: a cement space of 40 µm, and coping thickness of 500 µm. The prosthesis designed in the CAD software was saved as a CAD file. The CAD file was moved to milling equipment (Ceramill Motion 2; Amann Girrbach, Koblach, Austria) and a pre-sintered yttria-stabilized tetragonal zirconia polycrystalline block was milled (Ceramill Zolid, 1608000-23; Amann Girrbach, Koblach, Austria) to produce pre-sintered zirconia coping. Additionally, the zirconia coping was sintered according to the manufacturer’s recommendations. After sintering, the completed zirconia coping did not undergo any internal or external adjustments (Figure 2).

### 2.2. Marginal and Internal Fit Assessment

This study measured the marginal and internal fit of the prosthesis according to the five methods. Figure 3 shows a cross-sectional image obtained using the analysis guide template. The analysis guide template was used to obtain a cross-sectional image of the same part in each analysis method. Four regions (marginal gap, axial gap, angle gap, and occlusal gap) were measured on the four sides (buccal, lingual, mesial, and distal) of the prosthesis, and the gaps were measured at 10 points on each side (Figure 3). Thus, 40 points were measured for each region (marginal, axial, angle, and occlusal) (*n* = 40). A pilot experiment was conducted five times, and the calculation was made using a power analysis software (G*Power v3.1.9.2; Heinrich-Heine-Universität Düsseldorf) (Actual power = 97.6%; power = 97.5%; α = 0.05). All gap measurements were made by the same researcher. Since the researcher’s expertise could greatly impact the accuracy of the prosthesis fitness evaluation [48], assessments were conducted only after substantial training and practice.

#### 2.2.1. Cross-Sectional Method (CSM Group)

The CSM measurements were conducted as illustrated in Figure 4. Self-adhesive resin cement (RelyX U200; 3M ESPE, St. Paul, MN, USA) was mixed according to the manufacturer’s instructions and injected into the prosthesis. After adaptation to the abutment tooth, the prosthesis was pressed for ten minutes at a force of 10 Ncm, using a universal testing machine (Model 6022; Instron Co., Canton, MA, USA) since previous studies have shown that a force of 10 Ncm would not fracture the ceramic coping [38,49]. In addition, to avoid fracturing the zirconia coping while cutting, the sample was embedded using auto polymerizing acrylic resin (Orthodontic Resin; Dentsply Sirona, York, PA, USA). After the resin was cured, the sample was cut precisely into four pieces using a cutting machine (TechCut 4; Allied High-Tech Products, Compton, CA, USA) according to the analysis guide template. Images were then captured at 160 × magnification using an industrial video microscope system (IMS 1080P; SOMETECH, Seoul, Korea) (Figure 4A). The captured images were loaded into a software (ITPlus 5.0; SOMETECH, Seoul, Korea) to measure the gaps. Using the tools provided by the software, the desired positions were selected in the coping and abutment, and the gaps were measured.

#### 2.2.2. Silicone Replica Technique (SRT Group)

The silicone method is the same as the CSM, except for the injection of silicone into the coping and the cutting of the silicone replica (Figure 5). The zirconia coping was filled with a light body silicone (Aquasil Ultra XLV; Dentsply Detrey GmbH, Konstanz, Germany), and after adapting the crown on the study model, a pressure of 10 Ncm was applied using a universal testing machine. After polymerization of the light body silicone, the zirconia coping was removed and the light body silicone was covered with a medium body silicone (Aquasil Ultra Monophase; Dentsply Detrey GmbH, Konstanz, Germany) in order to enable stable cutting. After polymerization of the medium body silicone, the analysis guide template was combined with the study model. The silicone was cut using a razor blade (Personna; American Safety Razor, Staunton, VA, USA) according to the analysis guide template. Images were captured at 160 × magnification with an industrial video microscope system (IMS 1080P; SOMETECH, Seoul, Korea) (Figure 5A), which were then loaded in the software (ITPlus 5.0; SOMETECH, Seoul, Korea) to measure the gaps. Using the tools provided by the software, the desired positions were selected in the coping and abutment, and the gaps were measured.

#### 2.2.3. Triple Scan Method (TSM Group)

The TSM measurements were conducted as shown in Figure 6. This method measures the marginal and internal fit by superimposing the 3D scan data of the prosthesis and the abutment tooth based on the try-in status, and moving the 3D scan data to coordinates corresponding to the original position of the prosthesis and abutment. Three scan data for the TSM were obtained as follows: before cementation, the internal and external surfaces of the zirconia coping, as well as the study model, were scanned by a contact scanner (DS10; Renishaw plc, Gloucestershire, UK), and the data were saved in a standard tessellation language (STL). Afterward, the cemented zirconia coping and the study model were scanned and saved, as described above. The contact scanner, which was calibrated for accuracy, used a probe with a 0.5 mm diameter, contacting lightly and rising vertically at intervals of 200 µm. The data from the three scans were named the coping file, abutment file, and adaptation file. The analysis guide template was also scanned by an optical scanner (E1 scanner; 3Shape, Copenhagen, Denmark) and saved as an STL file.

The scan file was loaded into the analysis software (Geomagic Control X, 2018.0.1; 3D Systems, Rock Hill, SC, USA). To overlap the coping file and the abutment file, the adaptation file was set as reference data, and initial alignment followed by best-fit alignment was made. After the best-fit alignment, the overlap of the coping and the abutment files was confirmed at the accurate positions of the adaptation file. The adaptation file unnecessary for the measurement was deleted. In order to obtain cross-sectional images, the analysis guide template file was aligned to the abutment file and a virtual plane was set in the buccolingual and mesiodistal directions. In addition, the desired positions for gap measurements were selected on the set virtual plan, and the gaps were measured using Geomagic Control X (Figure 6A).

#### 2.2.4. Micro-Computed Tomography (MCT Group)

The MCT measurements were conducted as illustrated in Figure 7. In this method, the material density has a significant impact on the X-ray, and the boundaries between the different substances can be obscure. It is difficult for X-rays to penetrate various layers of cement and abutment for high-density materials such as zirconia [50]. Thus, this study performed MCT measurements twice. While rotating the sample located between the X-ray source and flat panel detector 360°, 400 sheets of 2D images were taken at an interval of 0.9°. First, a scan file of the abutment was obtained using Micro-CT (inspeXio SMX-225CT; Shimadzu, Kyoto, Japan) with a pixel matrix of 512 × 512, a tube voltage of 70 kV, a tube current of 50 μA, 1200 views, focal size of 4 µm, and slice thickness of 60 µm. The analysis guide template was then connected to the cemented zirconia coping and the study model, and another scan file was obtained in the same condition. The files scanned with Micro-CT were extracted as Tag Image File Format (TIFF) data. Using a 3D reconstruction software (VG Studio Max; Volume Graphics GmbH, Heidelberg, Germany), the TIFF data were converted into sliced images. When the images were reconstructed in Micro-CT data, the artifacts and noise were reduced through filtering. Three-dimensional images were obtained by scanning and overlapped twice on VG Studio Max software, and the zirconia coping and the abutment were saved as two 3D images, excluding the cement (Figure 7A).

In the case of images reconstructed to measure the gap, the cross-sectional images were obtained based on the analysis guide template (Figure 7B). The desired positions were selected on the VG Studio Max software and the gaps were measured.

#### 2.2.5. Optical Coherence Tomography (OCT Group)

The OCT measurements were conducted as illustrated in Figure 8. This study employed a swept-source (SS) OCT system (OCS1310V1; Thorlabs, Newton, NJ, USA) with a center wavelength of 1300 nm and bandwidth >97 nm. This SS-OCT system has an A-scan line rate of 100 kHz and sensitivity of 105 dB and supports a 12 mm imaging depth range at a sampling rate of 500 MS/s. The zirconia coping was installed in the accurate position of the abutment tooth and scanned in the buccolingual and mesiodistal directions in the SS-OCT system. The light generated from a laser source was divided into a reference arm and a sample arm. The light of the reference arm was reflected by the mirror, and that of the sample arm was reflected by the sample. An interference signal was generated by two reflected lights in a fiber coupler, obtained by a balanced photodetector, and then digitized using a data acquisition card. Finally, the digitized signal was reconstructed to a 2D image using an image processing software (Figure 8A) [51]. The scan range of the sample was set to 10 mm × 4 mm considering the sample size. The pixel size of the reconstructed 2D image was 1000 × 408 pixels. Since there is air between the zirconia coping and the abutment tooth, the index of refraction in an axial direction was set at 1.00 for the OCT image.

To evaluate the marginal and internal fit, the intensity peak profile was analyzed. Through coding in the MATLAB software (MathWorks; Natick, MA, USA), the intensity signal was detected in the direction of depth at the positions in which the gap measurements would be desired (Figure 8B). The intensity signal was obtained by designating a line at a position in which the gap measurement would be desired in a cross-sectional image and obtaining intensity values (between 0–255) at the pixel corresponding to the line. There was no reflected light between the zirconia coping and the abutment tooth as it was filled with air. Therefore, no interference signal was detected between the zirconia coping and the abutment tooth. Thus, the peak of the intensity signal provided coordinates using the data cursor of the MATLAB software, and the peak to peak of the intensity signal was measured by the gap between the zirconia coping and the abutment tooth (Figure 8B).

### 2.3. Statistical Analysis

All data were analyzed using SPSS statistical software (release 23.0; IBM, Chicago, IL, USA). First, the data were analyzed for normal distribution using the Shapiro–Wilk test, and since they did not show normal distribution, a Kruskal–Wallis H test was conducted (α = 0.05). Also, as a post-hoc test, the differences between the groups were analyzed using the Mann–Whitney U-test and Bonferroni correction method (α = 0.005).

## 3. Results

The values from the measurements for the marginal and internal fit are shown in Figure 9 and Table 1.

There were statistically significant differences in the fitness of the margin region among the five methods (*p* < 0.001). CSM (23.2 ± 5.3 µm) and SRT (33.5 ± 12.1 µm) did not differ significantly (*p* = 0.014) and showed the lowest value. Similarly, there were no statistically significant differences (*p* = 0.227) between TSM (74.1 ± 26.1 µm) and OCT (83.4 ± 22.1 µm), but they showed the highest values. Furthermore, MCT (45.9 ± 25.9 µm) did not have statistically significant differences compared to SRT (*p* = 0.090).

Some of the five methods showed statistically significant differences (*p* < 0.001) in the fitness of the axial region. CSM (83.7 ± 19.8 µm) and SRT (95.9 ± 52.9 µm) did not have statistically significant differences (*p* = 0.760) but demonstrated the highest values. On the other hand, although TSM (56.3 ± 30.1 µm), MCT (65.3 ± 47.7 µm), and OCT (44.8 ± 14.5 µm) did not have statistically significant differences (*p* > 0.005), they showed the lowest values.

Also, some of the five methods showed statistically significant differences (*p* < 0.001) in the fitness of the angle region. CSM (87.9 ± 17.2 µm) and MCT (84.3 ± 20.2 µm) did not have statistically significant differences (*p* = 0.651) and showed the lowest values. TSM (110.1 ± 13.9 µm) and OCT (118.2 ± 22.2 µm) also did not have statistically significant differences (*p* = 0.163), but they showed the highest values. Lastly, SRT (128.6 ± 17.3 µm) did not have statistically significant differences from OCT (*p* = 0.043).

There were some statistically significant differences (*p* < 0.001) in the fitness of the occlusal region between the five methods. CSM (125.4 ± 13.7 µm), SRT (140.5 ± 33.3 µm), TSM (120.3 ± 20.9 µm), and OCT (134 ± 18.9 µm) did not have statistically significant differences (*p* > 0.005), showing the highest values. On the other hand, MCT (102.6 ± 12.8 µm) showed the lowest value.

Figure 10 shows the difference of each method from CSM. Except for the angle region, the SRT showed the most approximate value.

## 4. Discussion

The present study compared five marginal and internal fit assessment methods for fixed prostheses. It was hypothesized that there would be no differences in the resulting values according to the five types of marginal and internal fit assessment methods. However, based on our findings, the null hypothesis was rejected (*p* < 0.001). The results of this study showed differences in the values obtained from the five types of marginal and internal fit assessment methods. In addition, there was a tendency to obtain similar marginal and internal fit measurements in CSM and SRT, and in TSM and OCT (Figure 9, Table 1).

In previous studies, in order to verify these differences, Oka et al. [14] produced a silicone replica by mixing a contrast medium with silicone and found that there were no significant differences in all the regions measured using MCT and SRT. On the other hand, the results of a comparison of MCT and SRT in our study did not show significant differences in the marginal and axial regions, but there were significant differences in the angle and occlusal regions in SRT with a high value. Han et al. [21] compared OCT and MCT and found a significant difference in OCT with a high value. Likewise, in our study, when OCT and MCT were compared, there was a significant difference in OCT with a high value except for the axial region. Boitelle et al. [39] compared the margin gap of SRT and that of TSM and found a significant difference in SRT with a high value. On the other hand, in our study, the margin gap of SRT compared with that of TSM showed a significant difference in SRT with a low value. The previous studies also showed different results based on the methods. The differences between our study and the previous studies are as follows. This study produced and applied a guide template to conduct analysis at the same position in the 2D image and made a comparison with the gap value of 40 points per region in a sample. The study was conducted with one sample to minimize production errors (e.g., study model, template, and coping) [52,53,54] and to only observe the differences between methods. In addition, the previous studies compared two methods [14,18,21,34,39,40], while this study compared the most possible methods used for assessing the fitness of the fixed prosthesis. Thus, through the results of our study, the differences in the result values based on the method could be compared.

Most previous studies have examined the fitness of the prosthesis according to the type of restoration material [11,36,44,55]. In addition, they mainly assessed if the marginal fit could be applied clinically [21,30]. Many previous studies regard a value of 100–120 µm as the clinically allowable range of the marginal fit [11,41,42,43] and recommend a range of 50–100 µm for the internal fit [23,56,57]. A poor marginal fit can increase plaque accumulation leading to secondary caries, periodontal disease, and endodontic inflammation [58]. Our study compared the marginal and internal fit methods, and according to the methods, the marginal fit was 23.2–83.4 µm (a 60.2 µm difference), and the internal fit (axial, angle, and occlusal) was 44.8–95.9 µm (a 51.1 µm difference), 84.3–128.6 µm (a 44.3 µm difference), and 102.6–140.5 µm (a 37.9 µm difference), respectively. If the marginal fit of a certain restoration material shows a clinical allowable result of about 60 µm using the SRT method, the result may deviate from the clinical allowable range using the OCT method. In contrast, even though the result is not in the clinical allowable range, it may be altered by changing the assessment method.

In previous studies, the CSM method [40,48] and SRT method [5,14,22,23] showed higher reliability than the other methods. However, many studies prefer the SRT method because it is a non-destructive and accurate method [22,23,26,30,59]. Our results of the marginal fit analysis by SRT were similar to that of CSM and MCT. Also, the results of the internal fit analysis by SRT were similar to CSM and MCT in the axial gap; OCT in the angle gap; and CSM and OCT in the occlusal gap. Overall, SRT shows a similar tendency to CSM. To obtain a cross-sectional image, excluding SRT and CSM, the samples should be digitized using the equipment specific to each method. Thus, the reason for the tendency to obtain similar values in SRT and CSM is probably an error in the process of digitizing the samples in each method. In the future, these differences will decrease with the development of better equipment and an improvement in the methods.

There were pros and cons with each method for assessing the fitness of prosthesis presented in this study (Table 2). From an economic point of view, OCT, MCT, and TSM require expensive equipment operated by experts along with the analysis software. SRT and CSM, on the other hand, have the merits of being relatively easy and low-cost methods [26,30]. The ability to perform an evaluation in the oral cavity is also an important factor for the assessment methods to be used in clinical trials. SRT and TSM can perform assessments in the oral cavity [53,60], while CSM, OCT, and MCT cannot be applied to the oral cavity. However, the OCT equipment can be developed for application as an intra-oral probe [61], as it is a non-radiological method.

Previous studies have shown that accessibility and various indications in clinical environments can have an impact on experimental methods. MCT has a disadvantage in that it is difficult for an X-ray to penetrate the various layers of cement and abutment for high-density materials like zirconia [50] (Figure 11). To overcome this, Oka et al. [14] produced a silicone replica, mixing a contrast medium with silicone, which was then imaged with MCT. On the other hand, our study obtained images of the abutment and the prosthesis after cementation and overlapped them with precision in the software. As a result, a clear boundary between the prosthesis and the abutment was obtained (Figure 7). OCT has the advantage that it could obtain a 3D image in real time [37], but no thick or opaque prosthesis could be applied due to the optical characteristics. Thus, there are pros and cons to each method, and these conditions should also be considered in the assessment.

In addition to the methods examined in our study, there are many fitness assessment methods that have been applied by others [14,28,30,40,48,60]. Also, the assessment methods used in various studies may differ depending on the environment of the laboratory. In addition, there is still no standard protocol to assesses the fitness of dental restorations [23]. The assessment methods used in varying conditions and the lack of standardization could lead to false interpretations and thus limit their comparison with the results of other studies. Likewise, the experimental conditions proposed in our study are not completely the same as those of the previous studies. However, our study made measurements using individual measuring devices, software, and by using methods proposed in previous studies in order to represent each assessment method. Therefore, it is recommended that future studies should compare our method with the methods proposed in other studies.

## 5. Conclusions

The null hypothesis of this study was rejected, and the five marginal and internal fit assessment methods showed different marginal and internal fit values. In our findings, there was a tendency of having similar marginal and internal fit in CSM and SRT, and in TSM and OCT. Therefore, the relatively simple and inexpensive SRT method can be an excellent alternative to CSM. According to the significance of these results, even if the assessment result values of the marginal and internal fit are in the clinical allowable range, the differences in the result values according to the method should be considered. A false interpretation of the marginal and internal fit assessments could result in inaccurate marginal fit values, which may make it difficult to perform long-term maintenance of the patient’s health. For this reason, it is necessary to establish a standard protocol for marginal and internal fit assessment of fixed prostheses.

## Figures and Tables

**Figure 1 jcm-08-00785-f001:**
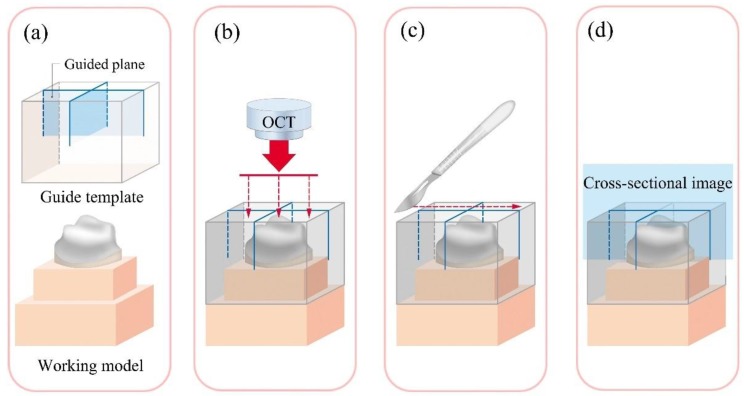
Acquisition and application of the same cross-sectional image. (**a**) Designed so that a guided plane would be formed according to the guide template and connected to the study model; (**b**) The optical beam in optical coherence tomography (OCT) was guided according to the template; (**c**) Cutting of the silicone replica in the silicone replica technique was guided according to the template; (**d**) All assessment methods were analyzed in an identical cross-sectional image according to the guide template.

**Figure 2 jcm-08-00785-f002:**
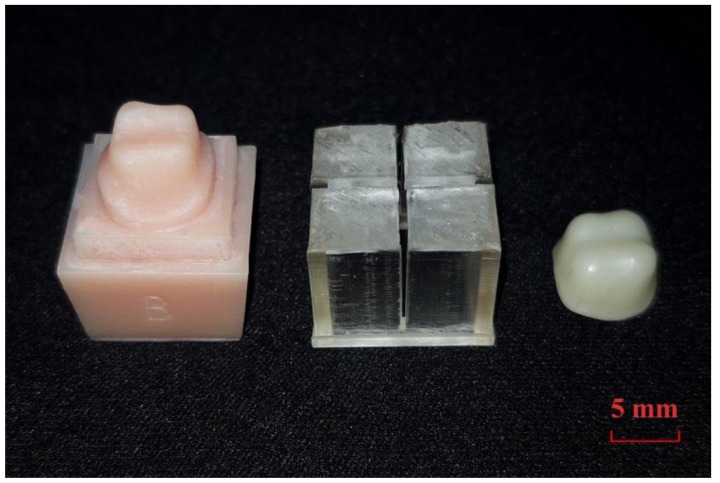
Study model, guide template, and zirconia coping from left to right.

**Figure 3 jcm-08-00785-f003:**
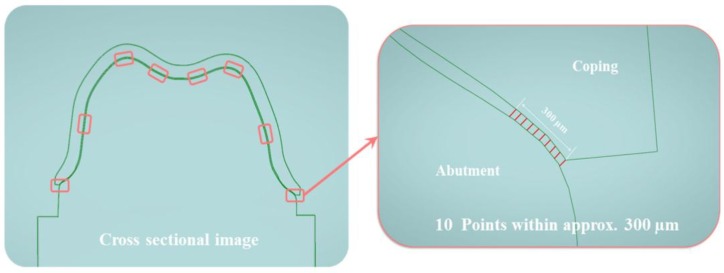
Position of analysis in the cross-sectional image. The right figure indicates the marginal fit and describes that 10 points have been measured within the range of about 300 µm.

**Figure 4 jcm-08-00785-f004:**
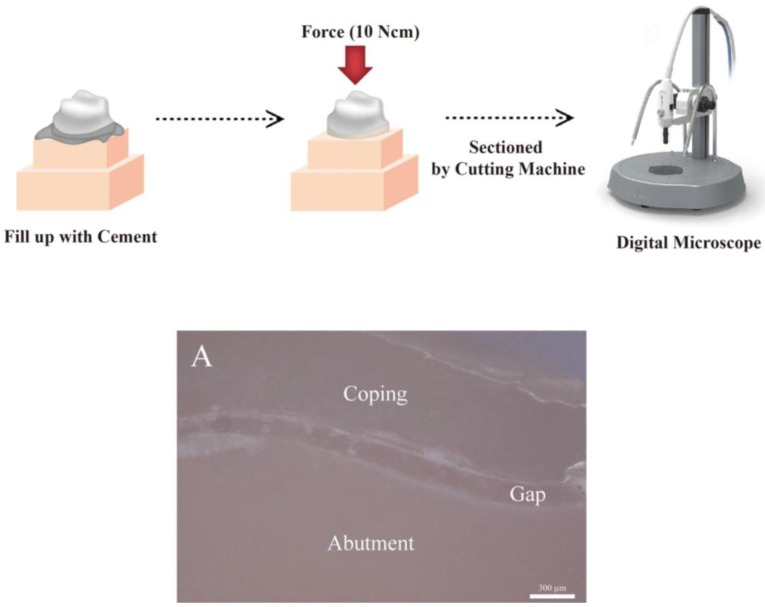
Schematic of the cross-sectional method (CSM). (**A**) Cross-sectional image of CSM for the occlusal gap.

**Figure 5 jcm-08-00785-f005:**
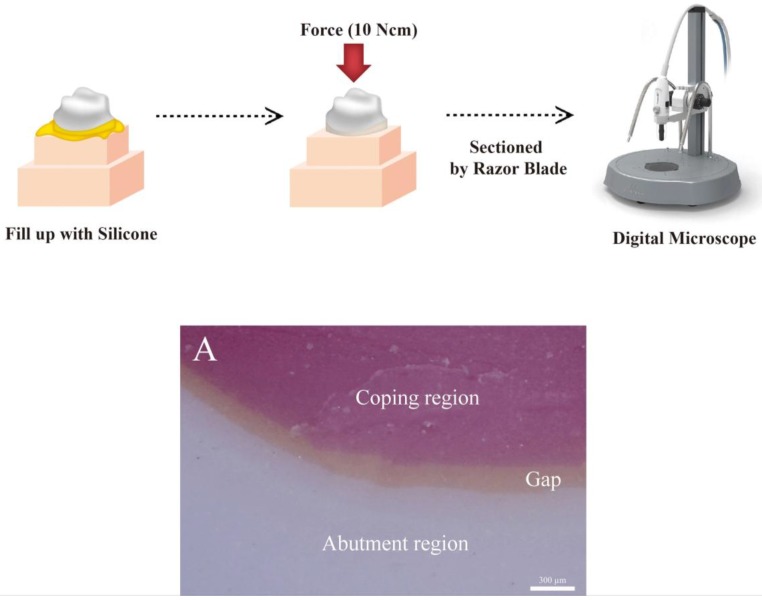
Schematic of the silicone replica technique (SRT). (**A**) Cross-sectional image of SRT for the occlusal gap.

**Figure 6 jcm-08-00785-f006:**
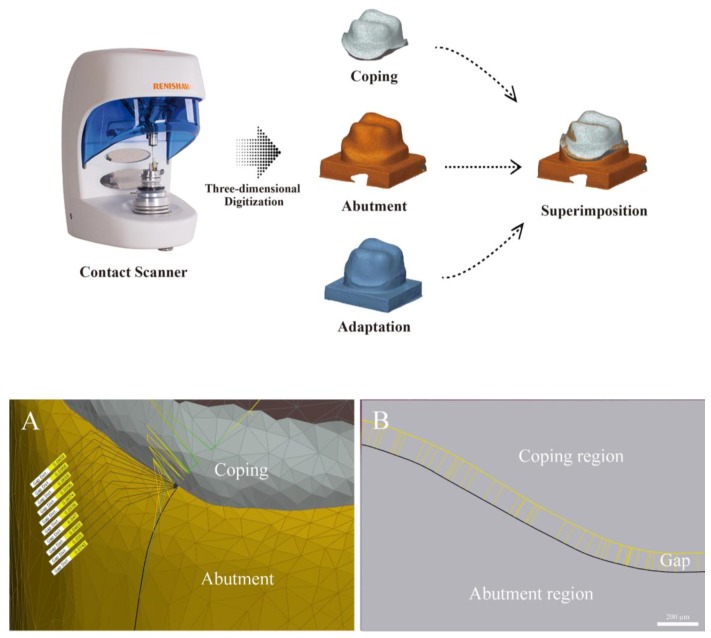
Schematic of the triple scan method (TSM). (**A**) Three-dimensional image; (**B**) cross-sectional image of the TSM for the occlusal gap.

**Figure 7 jcm-08-00785-f007:**
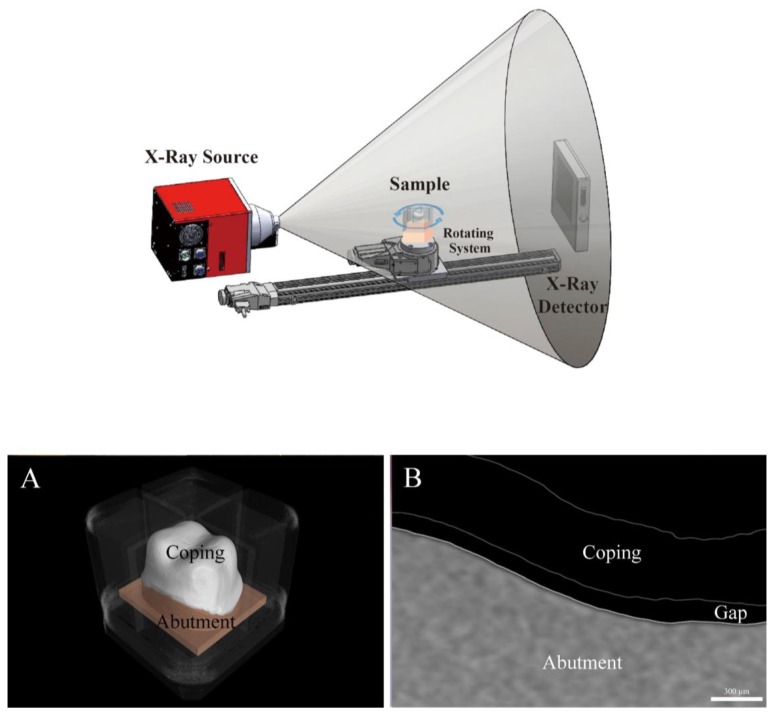
Schematic of micro-computed tomography (MCT) from the left. (**A**) Three-dimensional image obtained through MCT; (**B**) cross-sectional image of MCT for the occlusal gap.

**Figure 8 jcm-08-00785-f008:**
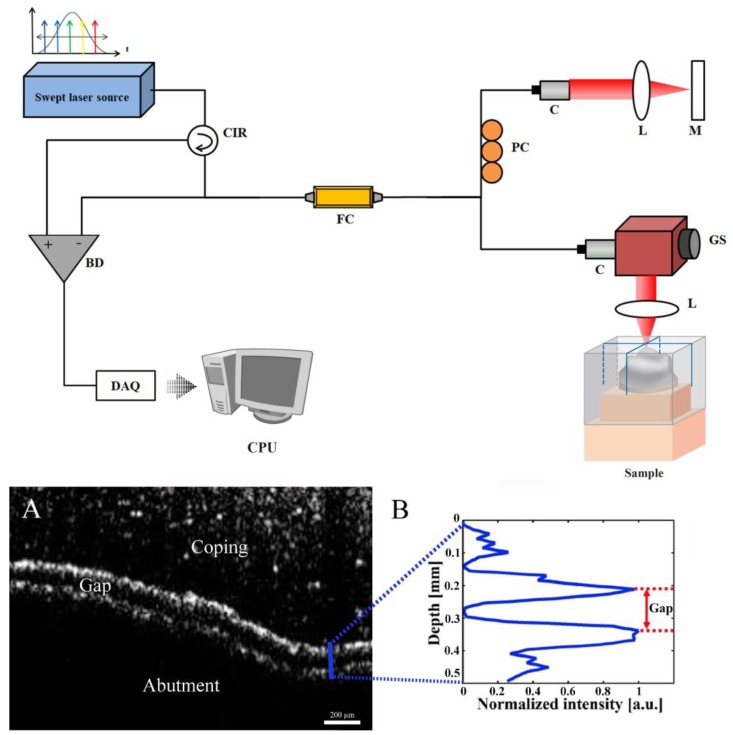
Schematic of the optical coherence tomography (OCT) from the left. (**A**) This indicates the method for measuring the occlusal region in the cross-sectional image of OCT; (**B**) in the intensity peak graph, the gap was calculated as the distance between the highest peak. BD: Balanced Detector, C: Collimator, CIR: Circulator, CPU: Central Processing Unit, DAQ: Data Acquisition, FC: Fiber Coupler, GS: Galvanometer Scanner, L: Lens, M: Mirror, PC: Polarization Controller.

**Figure 9 jcm-08-00785-f009:**
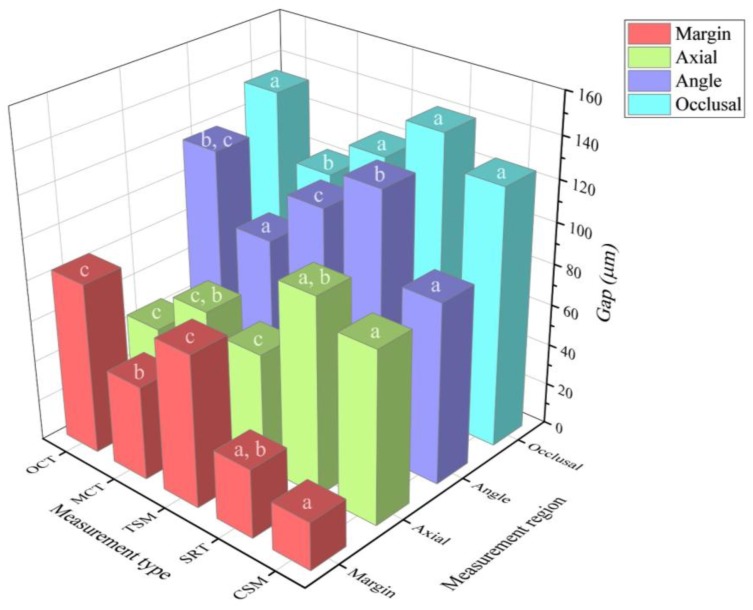
Result of the fitness measurement according to the measurement type and measurement region. Different superscript letters (**a**–**c**) indicate a statistically significant difference between the measurement types based on the Mann–Whitney U-test and Bonferroni correction method (*p* < 0.005).

**Figure 10 jcm-08-00785-f010:**
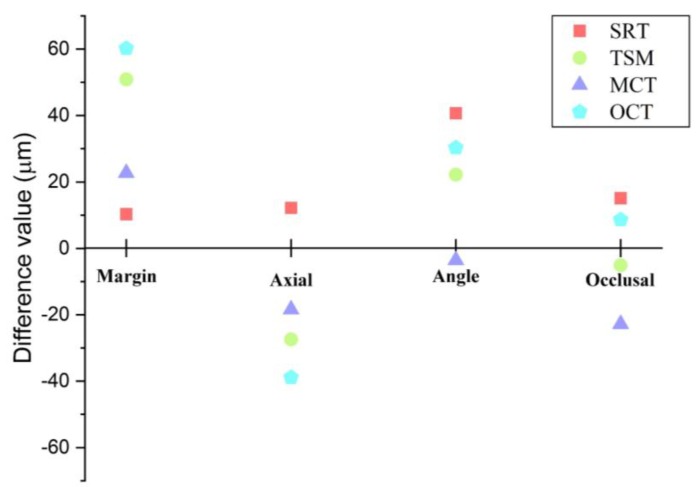
Differences in values compared to the cross-sectional method (CSM). The graph shows the difference from each method based on the CSM (baseline).

**Figure 11 jcm-08-00785-f011:**
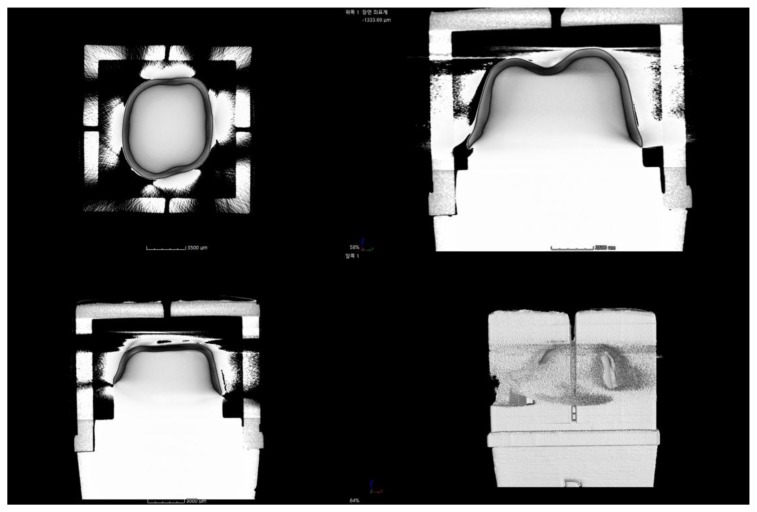
Image of the micro-computed tomography after cementation. Cross-sectional images show that marginal and internal gap measurements are not possible because the boundaries between the copings and abutments are uncertain.

**Table 1 jcm-08-00785-t001:** The marginal and internal gap of the cross-sectional method (CSM), silicone replica technique (SRT), triple scan method (TSM), micro-computed tomography (MCT), and optical coherence tomography (OCT). Gap values are depicted as mean (µm) ± standard deviation (SD).

Test method	Margin	Axial	Angle	Occlusal
CSM	23.2 ± 5.3 ^a^	83.7 ± 19.8 ^a^	87.9 ± 17.2 ^a^	125.4 ± 13.7 ^a^
SRT	33.5 ± 12.1 ^a,b^	95.9 ± 52.9 ^a,b^	128.6 ± 17.3 ^b^	140.5 ± 33.3 ^a^
TSM	74.1 ± 26.1 ^c^	56.3 ± 30.1 ^c^	110.1 ± 13.9 ^c^	120.3 ± 20.9 ^a^
MCT	45.9 ± 25.9 ^b^	65.3 ± 47.7 ^c,b^	84.3 ± 20.2 ^a^	102.6 ± 12.8 ^b^
OCT	83.4 ± 22.1 ^c^	44.8 ± 14.5 ^c^	118.2 ± 22.2 ^b,c^	134 ± 18.9 ^a^

Different superscript letters (a–c) indicate statistically significant differences among the test method groups based on the Mann–Whitney U-test and Bonferroni correction method (*p* < 0.005).

**Table 2 jcm-08-00785-t002:** Pros and cons with cross-sectional method (CSM), silicone replica technique (SRT), triple scan method (TSM), micro-computed tomography (MCT), and optical coherence tomography (OCT) in the presented study. Circle means good evaluation.

Pros and Cons	CSM	SRT	TSM	MCT	OCT
Use in the oral cavity	X	○	○	X	△
Economics	△	○	△	X	X
Accessibility in a clinical environment	△	○	○	X	X
Various indications	○	○	○	△	△
Reliability through previous studies	○	○	△	○	△

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
