# Peer review of "A Comparison Study of Marginal and Internal Fit Assessment Methods for Fixed Dental Prostheses"

_jcm, 2019, doi:10.3390/jcm8060785_

Round 1
Reviewer 1 Report
In the manuscript entitled: “A Comparison Study of Marginal and Internal Fit Assessment Methods for Fixed Dental Prostheses” the authors compared five marginal and internal fit assessment methods for fixed prostheses with a specially designed study model and analytical guide.
The authors evaluated the prosthetic’s marginal and internal fit according to the cross-sectional method (CSM), silicone replica technique (SRT), triple scan method (TSM), micro-computed tomography (MCT), and optical coherence tomography (OCT).
The authors found thatThe 5 methods showed significant differences in the 4 regions: the marginal, axial, angle, and occlusal regions showed low mean values. Internal fit was in the ranges of 44.8~95.9 μm (51.1 μm difference), 84.3~128.6 μm (44.3 μm difference), and 102.6~140.5 μm (37.9 μm difference), respectively
The authors concluded that marginal and internal fit showed significant differences depending on the method. Moreover, they stated that, despite the clinical allowable range shown in the assessment of the marginal and internal fit of the fixed prosthesis, the differences in the result values according to the method must be considered.
Major comments:
In general, the idea and innovation of this study, regards the analysis of of marginal and internal fit assessment methods for fixed dental prostheses is interesting, because the role of this adjuvant is validated but further studies on this topic could be an innovative issue in this field could be open an innovative matter of debate in literature by adding new information. Moreover, there are few reports in the literature that studied this interesting topic with this kind of study design.
The study was well conducted by the authors; However, there are some concerns to revise that are described below.
The introduction section resumes the existing knowledge regarding the importance of the prosthesis fit.
However, as the importance of the topic, the reviewer strongly recommends to update the literature through read, discuss and cites in the references with great attention all of those recent interesting articles, that helps the authors to better introduce and discuss the aim of the study in light of the some factors which influences the marginal fit in the periodontium: 1) Ariganello MB, Guadarrama Bello D, Rodriguez-Contreras A, Sadeghi S, Isola G, Variola F, Nanci A. Surface nanocavitation of titanium modulates macrophage activity. Int J Nanomedicine. 2018;13:8297-8308. doi: 10.2147/IJN.S185436. 2) Hagenfeld D, Mutters NT, Harks I, Koch R, Kim TS, Prehm P. Hyaluronan-mediated mononuclear leukocyte binding to gingival fibroblasts. Clin Oral Investig. 2018 Mar;22(2):1063-1070. 3) Matarese G, Ramaglia L, Fiorillo L, Cervino G, Lauritano F, Isola G. Implantology and Periodontal Disease: The Panacea to Problem Solving? Open Dent J. 2017 Aug 30;11:460-465. doi: 10.2174/1874210601711010460.
The authors should be better specified, at the end of the introduction section, the rational of the study and the relative null-hypothesis. In the material and methods section, should better clarify the preparation of the bloks and the triple scan method.
The discussion section appears well organized with the relevant paper that support the conclusions, even if the authors should better discuss the relationship between marginal gap and periodontal disease. The conclusion should reinforce in light of the discussions.
In conclusion, I am sure that the authors are fine clinicians who achieve very nice results with their adopted protocol. However, this study, in my view does not in its current form satisfy a very high scientific requirement for publication in this journal and requests a revision before publication.
Minor Comments:
Abstract:
- Better formulate the introduction section by better describe the background
Introduction:
- Page 2, Line 50: please add the relative sentence
Discussion
- Please add a specific sentence that clarifies the results obtained in the first part of the discussion
- Page 12 last paragraph of discussion: Please reorganize this paragraph that is not clear
Author Response
We are grateful to the reviewers for their critical comments and useful suggestions that have helped us to greatly improve our paper. As indicated in the following responses, we have reflected on all these comments in the revised version of our paper. Furthermore, we have had the manuscript checked by a professional English editing service.
Reviewer #1
Major comments:
1. The introduction section resumes the existing knowledge regarding the importance of the prosthesis fit.
However, as the importance of the topic, the reviewer strongly recommends to update the literature through read, discuss and cites in the references with great attention all of those recent interesting articles, that helps the authors to better introduce and discuss the aim of the study in light of the some factors which influences the marginal fit in the periodontium.
Response: Thank you for your suggestion for improving the quality of the manuscript. We have carefully considered your comments. As the reviewer notes, we have added text and references to improve the Introduction section.
P.1, lines 38-40: “On the other hand, a poor marginal fit has a negative impact on the periodontium, making it difficult to perform long-term maintenance of the patient's health following implant placement [5,6,7].”
5. Ariganello, M.B.; Guadarrama Bello, D.; Rodriguez-Contreras, A.; Sadeghi, S.; Isola, G.; Variola, F.; Nanci, A. Surface nanocavitation of titanium modulates macrophage activity. Int. J. Nanomedicine. 2018, 13, 8297-8308. https://doi.org/10.2147/IJN.S185436.
6. Hagenfeld, D; Mutters, N.T.; Harks, I.; Koch, R.; Kim, T.S.; Prehm, P. Hyaluronan-mediated mononuclear leukocyte binding to gingival fibroblasts. Clin. Oral. Investig. 2018, 22, 1063-1070. https://doi.org/10.1007/s00784-017-2188-x.
7. Matarese, G.; Ramaglia, L.; Fiorillo, L.; Cervino, G.; Lauritano, F.; Isola, G. Implantology and Periodontal Disease: The Panacea to Problem Solving? Open Dent. J. 2017 30, 11, 460-465. https://doi.org/10.2174/1874210601711010460.
2. The authors should be better specified, at the end of the introduction section, the rational of the study and the relative null-hypothesis.
Response: We agree with the reviewer’s suggestion. We have specifically addressed the issue that you point out.
P.2, lines 93-99: “….Thus, for precise measurement, it is necessary to verify the differences among the various methods recently presented.
In this study, in order to verify the difference through objective comparison of five evaluation methods, the marginal and internal fit were measured in cross-sectional images obtained from the sample under identical conditions. This study aimed to conduct a comparative assessment of the marginal and internal fit methods of the fixed prosthesis introduced in the previous studies (CSM, SRT, TSM, MCT, and OCT). A null hypothesis was set as follows: there would be no differences in the marginal and internal (axial, angle, and occlusal region) fit measured according to the CSM, SRT, TSM, MCT, and OCT assessment methods.”
3. In the material and methods section, should better clarify the preparation of the bloks and the triple scan method.
Response: We very much appreciate the reviewer’s comment and respect the reviewer’s insight. We have specifically addressed the issue you highlighted.
P.5, lines 161-162: “In addition, to avoid fracturing the zirconia coping while cutting, a block was produced, embedding it in the sample was embedded using auto polymerizing acrylic resin (Orthodontic Resin; Dentsply Sirona, York, PA, USA).”
P.6, lines 193-196: “The TSM measurements were conducted as shown in Figure 6. ]This method measures the marginal and internal fit by superimposing the 3D scan data of the prosthesis and the abutment tooth based on the try-in status, and moving the 3D scan data to coordinates corresponding to the original position of the prosthesis and abutment. Three scan data for TSM were obtained as follows….”
4. The discussion section appears well organized with the relevant paper that support the conclusions, even if the authors should better discuss the relationship between marginal gap and periodontal disease. The conclusion should reinforce in light of the discussions.
Response: Thank you for your suggestion for improving the quality of the manuscript. We have carefully considered your comments. We have added the relationship between marginal gap and periodontal disease in the discussion section, and have reinforced the conclusion.
P.12, lines 347-349: “…. Many previous studies regard a value of 100–120 µm as the clinical allowable range of the marginal fit [11,41–43] and recommend a range of 50–100 µm for the internal fit [23,56,57]. A poor marginal fit can increase plaque accumulation leading to secondary caries, periodontal disease, and endodontic inflammation [58]. Our study compared the marginal and internal fit methods, and according to the methods, the marginal fit was ….”
58. El Ghoul, W.A.; Özcan, M.; Ounsi, H.; Tohme, H.; Salameh, Z. Effect of different CAD-CAM materials on the marginal and internal adaptation of endocrown restorations: An in vitro study. J. Prosthet. Dent. 2019. https://doi.org/10.1016/j.prosdent.2018.10.024.
P.14, lines 403-412: “The null hypothesis of this study was rejected, and the 5 marginal and internal fit assessment methods showed different marginal and internal fit values. In our findings, there was a tendency of having similar marginal and internal fit in CSM and SRT, and in TSM and OCT. Therefore, the relatively simple and inexpensive SRT method can be an excellent alternative to the CSM. According to the significance of these results, even if the assessment result values of the marginal and internal fit are in the clinical allowable range, the differences in the result values according to the method should be considered. False interpretation of the marginal and internal fit assessments could lead to inaccurate marginal fit values, which may make it difficult to perform long-term maintenance of the patient's health. For this reason, it is necessary to establish a standard protocol for marginal and internal fit assessment of fixed prostheses.”
Minor comments:
1. Abstract:
- Better formulate the introduction section by better describe the background.
Response: We agree with the reviewer’s suggestion. We have better described the background in the abstract section.
P.1, lines 18-20: “Numerous studies have previously evaluated the marginal and internal fit of fixed prostheses; however, few reports have performed an objective comparison of the various methods used for their assessment. The purpose of this study was to compare five marginal and internal fit assessment methods for fixed prostheses….”
2. Introduction:
- Page 2, Line 50: please add the relative sentence.
Response: Thank you for your comment. We have added the sentence.
P.2, lines 48: “CSM is a method in which after cementation, the desired part of the prosthesis is cut and measured with an optical or electronic microscope. Since the actual prosthesis is cut and measured, it has its merits in that it can measure the accurate internal and marginal fit of the prosthesis; however, it is disadvantageous in that samples should be destroyed, and measurement cannot be made directly in the oral cavity [8,9,26].”
3. Discussion
- Please add a specific sentence that clarifies the results obtained in the first part of the discussion.
Response: Thank you for your suggestion for improving the quality of the manuscript. We have specifically addressed the issue that you point out.
P.13, lines 319-322: “The present study compared five marginal and internal fit assessment methods for fixed prostheses. It was hypothesized that there would be no differences in the resulting values according to the 5 types of marginal and internal fit assessment methods. However, based on our findings, the null hypothesis was rejected (p < 0.001). In the results of this presented study, there were difference in result values according to the 5 types of marginal and internal fit assessment methods. In addition, there was a tendency of having similar marginal and internal fit in CSM and SRT, and in TSM and OCT (Figure 9, Table 1).”
4. Discussion
- Page 12 last paragraph of discussion: Please reorganize this paragraph that is not clear.
Response: Thank you for your suggestion for improving the quality of the manuscript. We have reorganized the text to improve the discussion section. We have also added a table so that the text is more readily comprehensible to a reader.
P.14, lines 375-388:
“There were pros and cons with each method for assessing the fitness of prosthesis in the presented in this study (Table 2). From an economic point of view, OCT, MCT, and TSM require expensive equipment operated by experts, along with a software. SRT and CSM, on the other hand have the merits of being relatively easy and low-cost methods [26,30]. The ability to perform evaluation in the oral cavity is also an important factor for the assessment methods to be used in clinical trials. SRT and TSM can perform assessments in the oral cavity [53,60], while CSM, OCT, and MCT cannot be applied to the oral cavity. However, the OCT equipment can be developed for application as an intra-oral probe [61] since it is a non-radiological method.
Previous studies have shown that accessibility and various indications in clinical environments can have an impact on the experimental methods. MCT has a disadvantage in that it is difficult for an x-ray to penetrate the various layers of cement and abutment for high-density materials like zirconia [50] (Figure 10). To overcome this, Oka et al. [14] produced a silicone replica, mixing a contrast medium with silicone, which was then imaged with MCT. On the other hand, our study obtained images of the abutment and the prosthesis after cementation, and overlapped them with precision in the software. As a result, a clear boundary between the prosthesis and the abutment was obtained (Figure 7). OCT has the advantage that it could obtain a 3D image in real time [37], but no thick or opaque prosthesis could be applied due to the optical characteristics. Thus, there are pros and cons with each method, and these conditions should also be considered in the assessment.”
Table 2. Pros and cons with cross-sectional method (CSM), silicone replica technique (SRT), triple scan method (TSM), micro-computed tomography (MCT), and optical coherence tomography (OCT) in the presented study. Circle means good evaluation.
Pros and cons | CSM | SRT | TSM | MCT | OCT |
Use in the oral cavity | X | ○ | ○ | X | △ |
Economics | △ | ○ | △ | X | X |
Accessibility in a clinical environment | △ | ○ | ○ | X | X |
Various indications | ○ | ○ | ○ | △ | △ |
Reliability through previous studies | ○ | ○ | △ | ○ | △ |
Reviewer 2 Report
Dear Editor,
thank you for giving me the opportunity to retire this interesting paper.
The topic is very original and the research is well done; moreover there are some minor points which must be improved.
-Abstract: it is too long (more than 200 words allowed) and it must be reduced.
-Keywords: I suggest to change "Cross-sectional method", "Silicone replica technique" and "Triple scan 38 method" with "Dentistry", "Dental prosthesis" and "Zirconia crown" for better focusing the topic of the research.
-English style must be improved, I suggest to use the English editing service. Particularly, there are many too short sentences which may be linked. I.E.
The marginal fit of a fixed prosthesis is one of the most important elements for successful prosthetic treatment [1,2] because an ideal marginal fit maintains healthy periodontal status and prevents cement dissolution [3,4]. In addition, excellent internal fit increases horizontal and vertical forces in prosthesis-eliminating resistance [5] and, for these reasons, many studies are conducted on the marginal and internal fit to determine the prognosis of a prosthesis [1–35].
-Conclusion: I suggest to use a discorsive style instead of following a "points discussion".
Best regards
Author Response
We are grateful to the reviewers for their critical comments and useful suggestions that have helped us to greatly improve our paper. As indicated in the following responses, we have reflected all these comments in the revised version of our paper. Furthermore, we have had the manuscript checked by a professional English editing service.
Reviewer #2
1. Abstract: it is too long (more than 200 words allowed) and it must be reduced.
Response: We agree with the reviewer’s suggestion. We have reduced the abstract section to 200 words.
2. Keywords: I suggest to change "Cross-sectional method", "Silicone replica technique" and "Triple scan 38 method" with "Dentistry", "Dental prosthesis" and "Zirconia crown" for better focusing the topic of the research.
Response: Thank you for your comment. We have revised the keywords.
P.1, lines 32: “Marginal and internal fit; Dentistry; Dental prosthesis; Zirconia crown; Micro-computed tomography; Optical coherence tomography”
3. English style must be improved, I suggest to use the English editing service. Particularly, there are many too short sentences which may be linked. I.E.
The marginal fit of a fixed prosthesis is one of the most important elements for successful prosthetic treatment [1,2] because an ideal marginal fit maintains healthy periodontal status and prevents cement dissolution [3,4]. In addition, excellent internal fit increases horizontal and vertical forces in prosthesis-eliminating resistance [5] and, for these reasons, many studies are conducted on the marginal and internal fit to determine the prognosis of a prosthesis [1–35].
Response: Thank you for your suggestion for improving the quality of the manuscript. We have had the manuscript checked by a professional English editing service.
4. Conclusion: I suggest to use a discorsive style instead of following a "points discussion".
Response: We very much appreciate the reviewer’s comment and respect the reviewer’s insight. We have revised the issue that you point out.
P.15, lines 404-413: “The null hypothesis of this study was rejected, and the 5 marginal and internal fit assessment methods showed different marginal and internal fit values. In our findings, there was a tendency of having similar marginal and internal fit in CSM and SRT, and in TSM and OCT. Therefore, the relatively simple and inexpensive SRT method can be an excellent alternative to the CSM. According to the significance of these results, even if the assessment result values of the marginal and internal fit are in the clinical allowable range, the differences in the result values according to the method should be considered. False interpretation of the marginal and internal fit assessments could result in inaccurate marginal fit values, which may make it difficult to perform long-term maintenance of the patient's health. For this reason, it is necessary to establish a standard protocol for marginal and internal fit assessment of fixed prostheses.”
Reviewer 3 Report
I read this paper with great attention.
My comments are as follows:
The study compares five very different methods. As you say in line75-80, previous studies compared only two methods. In my opinion, this was a proper approach, as it is almost impossible to get a coherent result when comparing such different methods as you did. Probably, this is the reason your results showed such great differences. It is obvious your null hypothesis did not confirm.
The introduction part is messy, it has to be structured in a more coherent way.
In the Materials and Methods part you mention that your experiments were conducted referring to the methods of previous studies. In my opinion you should briefly explain these methods, maybe not all readers are interested in consulting your references and further read the previous studies you are referring at.
line 19-20 "To compare five marginal and internal fit assessment methods for fixed prostheses.
Specially designed study model and analytical guide were produced for this study. The sample was..." This has to be a mistake, it is not understandable. Please rewrite in a proper manner.
line 43-44 "excellent internal fit increases horizontal and vertical forces in prosthesis-eliminating resistance". Sorry, not understandable, again.
line 95-96 "The study model was produced in special designs, and a guide template for analysis was produced to compare the methods for assessing the marginal and internal fit of the prosthesis." Please explain what do you mean by this.
line 97 "To produce a study model, in the study model..." This is not a proper expression. Please rephrase.
line 109-111. Please rephrase. Not understandable.
line 121-122 "The designed prosthesis was moved to zirconia milling equipment..." How did you move the designed prosthesis (in fact a CAD file) to the milling device?! Please rewrite this.
Please consider dividing Figure 4 in two separate Figures. Same for Figure 5, 6, 7. What do you mean by "from the left" in the figures caption?
line 309 "In the results of this study, different results were observed" Please rephrase.
line 310-311. "In addition, there was a tendency of having similar marginal and internal
fit depending on the method." I do not understand what you mean by this.
In my opinion, the entire structure of the paper has to be rethought and the article rewritten.
Author Response
We are grateful to the reviewers for their critical comments and useful suggestions that have helped us to greatly improve our paper. As indicated in the following responses, we have reflected all these comments in the revised version of our paper. Furthermore, we have had the manuscript checked by a professional English editing service.
Reviewer #3
1. The study compares five very different methods. As you say in line75-80, previous studies compared only two methods. In my opinion, this was a proper approach, as it is almost impossible to get a coherent result when comparing such different methods as you did. Probably, this is the reason your results showed such great differences. It is obvious your null hypothesis did not confirm.
Response: We appreciate the reviewer’s comment very much and respect the reviewer’s insight. We have carefully considered your comments. As the reviewer suggested, we have restructured the entire paper. We believe that most of the points made by reviewer 1, 2, and 3’s comments have been resolved.
Like the reviewer’s comment, we also considered previous studies comparing only two methods to be the proper approach. On the other hand, previous studies by Han et al. [18], Boitelle et al. [36], and Zeller et al. [37] showed differences in the results, even though only two methods were compared. Thus, we truly thought the differences in the result values were based on the assessment methods, with reference to previous studies. Additionally, in previous studies the comparisons was conducted in order to increase the reliability of the assessment method, and the focus of the manuscripts were not to compare the assessment methods. The purpose of our study was to compare the measured values based on the assessment methods presented in the previous studies.
The results of the presented study showed numerous differences based on 5 assessment methods. Currently, many assessment methods are being applied to evaluate the fitness of the prosthesis, and the findings presented in our study suggest that the differences based on the assessment methods should be considered.
We have revised the manuscript so that this study is readily comprehensible to readers and we have addressed the issue that you point out.
2. The introduction part is messy, it has to be structured in a more coherent way.
Response: Thank you for your suggestion for improving the quality of the manuscript. We have revised the introduction section in a more coherent way.
3. In the Materials and Methods part you mention that your experiments were conducted referring to the methods of previous studies. In my opinion you should briefly explain these methods, maybe not all readers are interested in consulting your references and further read the previous studies you are referring at.
Response: We agree with the reviewer’s suggestion. We have deleted the issue that you point out.
4. line 19-20 "To compare five marginal and internal fit assessment methods for fixed prostheses.
Specially designed study model and analytical guide were produced for this study. The sample was..." This has to be a mistake, it is not understandable. Please rewrite in a proper manner.
Response: Thank you for your accurate comment. We have revised the issue that you point out so that the abstract is readily comprehensible to readers.
P.1, lines 21: “….Specially designed study model and analytical guide were produced for this study. The specially designed sample was....”
5. line 43-44 "excellent internal fit increases horizontal and vertical forces in prosthesis-eliminating resistance". Sorry, not understandable, again.
Response: Thank you for your comments to improving the quality of the paper. We have revised the issue that you point out so that the sentence is readily comprehensible to readers.
P.1, lines 40: “In addition, excellent internal fit increases horizontal and vertical retention force of prosthesis in prosthesis-eliminating resistance [5].”
6. line 95-96 "The study model was produced in special designs, and a guide template for analysis was produced to compare the methods for assessing the marginal and internal fit of the prosthesis." Please explain what do you mean by this.
Response: Thank you for your comments to improving the quality of the paper. We have revised the issue that you point out so that the sentence is readily comprehensible to readers.
P.3, lines 103-104: “The study model and the guide template were specially designed, and a guide template for analysis was produced to compare the methods for assessing the marginal and internal fit of the prosthesis (Figure 1).”
7. line 97 "To produce a study model, in the study model..." This is not a proper expression. Please rephrase.
Response: Thank you for your comments. We have revised the issue that you point out so that the sentence is readily comprehensible to readers.
P.3, lines 104: “To produce a study model, in In the study dental model (ANA-4; Frasaco GmbH, Tettnang, Germany), the upper right first molar was prepared as follows: 1.5 mm of occlusal surface reduction, 1 mm of axial wall reduction, a 1.2 mm chamfer margin, and a 6-degrees of vertical angle.”
8. line 109-111. Please rephrase. Not understandable.
Response: Thank you for your accurate comment. We have rewritten the sentence that you point out
P.3, lines 116-119: “Resins were used for 3D printing the model (ZMD-1000B, ZMD0171208B02; Dentis, Daegu, Republic of Korea), as well as the guide (ZMD-1000B, ZCL0170621A06; Dentis, Daegu, Republic of Korea).”
9. line 121-122 "The designed prosthesis was moved to zirconia milling equipment..." How did you move the designed prosthesis (in fact a CAD file) to the milling device?! Please rewrite this.
Response: Thank you for your accurate comment. We have rewritten the sentence that you point out
P.4, lines 129-130: “The prosthesis designed in the CAD software was saved as a CAD file. The designed prosthesis CAD file was moved to zirconia a milling equipment (Ceramill Motion 2; Amann Girrbach, Koblach, Austria) and ….”
10. Please consider dividing Figure 4 in two separate Figures. Same for Figure 5, 6, 7. What do you mean by "from the left" in the figures caption?
Response: We appreciate the reviewer’s comment and respect the reviewer’s insight. We have divided Figure 4 in two separate Figures. We have deleted the sentence ("from the left") that you point out.
11. line 309 "In the results of this study, different results were observed" Please rephrase.
Response: Thank you for your comments. We have revised the issue that you point out so that the sentence is readily comprehensible to readers.
P.12, lines 321-322: “The results of this study showed differences in the values obtained from the 5 types of marginal and internal fit assessment methods. observed according to the marginal and internal fit method used.”
12. line 310-311. "In addition, there was a tendency of having similar marginal and internal fit depending on the method." I do not understand what you mean by this.
Response: Thank you for your comments. We have revised the issue that you point out so that the sentence is readily comprehensible to readers.
P.12, lines 323-324: “In addition, there was a tendency to obtain similar marginal and internal fit measurements in CSM and SRT, and in TSM and OCT (Figure 9, Table 1).”
Round 2
Reviewer 1 Report
In the R1 version of the manuscript entitled: “A Comparison Study of Marginal and Internal Fit Assessment Methods for Fixed Dental Prostheses” the authors followed all the issues suggested by the reviewer. Though the changes based on the reviewer comments, almost of the criticisms were carefully analysed and solved.
I have carefully evaluated all parts of the manuscript. I believe that the article, in this version, is now adequate for publication in this journal.
Reviewer 3 Report
Thank you for your answers.I have no further comments and suggestions.